# Heme-induced genes facilitate endosymbiont (*Sodalis glossinidius*) colonization of the tsetse fly (*Glossina morsitans*) midgut

Laura J. Runyen-Janecky[1]*, Jack D. Scheutzow[1], Ruhan Farsin[1], Leah F. Cabo[1], Katie E. Wall[1], Katrina M. Kuhn[1], Rashel Amador[1], Shaina J. D'Souza[1], Aurelien Vigneron[2¤], Brian L. Weiss [2]*

1 Department of Biology, University of Richmond, Richmond, Virginia, United States of America,
2 Department of Epidemiology of Microbial Diseases, Yale School of Public Health, New Haven, Connecticut, United States of America

¤ Current address: Univ Lyon, Université Claude Bernard Lyon 1, CNRS, INRAE, VetAgro Sup, UMR Ecologie Microbienne, Villeurbanne, France
* lrunyenj@richmond.edu (LJR); brian.weiss@yale.edu (BLW)

**Data Availability Statement:** All relevant data are within the manuscript and its Supporting Information files.

## Abstract

Tsetse flies (*Glossina* spp.) feed exclusively on vertebrate blood. After a blood meal, the enteric endosymbiont *Sodalis glossinidius* is exposed to various environmental stressors including high levels of heme. To investigate how *S. glossinidius morsitans* (*Sgm*), the *Sodalis* subspecies that resides within the gut of *G. morsitans*, tolerates the heme-induced oxidative environment of tsetse's midgut, we used RNAseq to identify bacterial genes that are differentially expressed in cells cultured in high versus lower heme environments. Our analysis identified 436 genes that were significantly differentially expressed (> or < 2-fold) in the presence of high heme [219 heme-induced genes (HIGs) and 217 heme-repressed genes (HRGs)]. HIGs were enriched in Gene Ontology (GO) terms related to regulation of a variety of biological functions, including gene expression and metabolic processes. We observed that 11 out of 13 *Sgm* genes that were heme regulated *in vitro* were similarly regulated in bacteria that resided within tsetse's midgut 24 hr (high heme environment) and 96 hr (low heme environment) after the flies had consumed a blood meal. We used intron mutagenesis to make insertion mutations in 12 *Sgm* HIGs and observed no significant change in growth *in vitro* in any of the mutant strains in high versus low heme conditions. However, *Sgm* strains that carried mutations in genes encoding a putative undefined phosphotransferase sugar (PTS) system component (SG2427), fucose transporter (SG0182), bacterioferritin (SG2280), and a DNA-binding protein (SGP1-0002), presented growth and/or survival defects in tsetse midguts as compared to normal *Sgm*. These findings suggest that the uptake up of sugars and storage of iron represent strategies that *Sgm* employs to successfully reside within the high heme environment of its tsetse host's midgut. Our results are of epidemiological relevance, as many hematophagous arthropods house gut-associated bacteria that mediate their host's competency as a vector of disease-causing pathogens.

**Funding:** This study was supported by NIH award AI124105 to LRJ. We also thank Dr. Serap Aksoy (Yale School of Public Health) for providing financial support required to rear the tsetse flies used in this study. The funders had no role in the study design, data collection and analysis, decision to publish, or preparation of the manuscript.

**Competing interests:** The authors have declared that no competing interests exist.

## Author summary

Tsetse flies feed exclusively on vertebrate blood. This nutrient source contains large quantities of heme, which can be toxic to the fly's associated microorganisms. We investigated the genetic mechanisms that underlie the ability of the bacterial endosymbiont, *Sodalis glossinidius* (*Sgm*), to successfully reside within tsetse's heme-rich midgut. Exposure of cultured *Sgm* to high levels of heme induced changes in the expression of genes that encode proteins involved in transcription, replication and repair of DNA, inorganic ion transport, and carbohydrate transport and metabolism processes. Changes in the expression of several of these same *Sgm* genes also occurred within tsetse's midgut following exposure to a blood meal. *Sgm* genetically engineered to present mutations in several of these heme regulated genes were unable to successfully colonize tsetse's gut. Our results provide insight into how bacteria that live in the gut of blood feeding arthropods mitigate the toxic effects of excessive heme. This information is of epidemiological relevance, as many of these bacteria influence their host's ability to transmit disease pathogens that cause disease in humans and domesticated animals.

## Introduction

Several arthropod taxa use vertebrate blood as their sole source of nutrients or as a nutrient supplement during metabolically costly reproductive processes. These arthropods are of epidemiological importance because they can transmit disease-causing pathogens between vertebrate hosts when they feed. In addition to potentially harboring pathogens, many hematophagous arthropods house a population-dependent assortment of symbiotic bacteria in their midgut that comes into direct contact with vertebrate blood each time their host feeds. These arthropods, pathogens, and symbiotic bacteria all face the unique challenge of accessing the rich supply of iron in the blood while overcoming the toxic effects of abundant free heme present in the environment during digestion of the meal. Meeting this challenge is vitally important for the survival of these organisms, which have presumably evolved several heme tolerance and detoxification mechanisms.

Heme (ferri-protoporphyrin IX) is composed of a heterocyclic organic porphyrin ring covalently bound to one ferrous iron atom. The majority of heme in the human body (~67%) is in hemoglobin, which is predominantly found in erythrocytes at a concentration of ~10 mM [1, 2]. Hematophagous arthropods ingest large amounts of vertebrate blood at each feeding. During digestion, heme, and the iron bound to it, are released into the midgut in large quantities, at which point they become toxic [1]. Specifically, free iron released from heme catalyzes the formation of hydroxyl radicals via Fenton chemistry, which directly damages a variety of biomolecules (e.g. DNA, proteins). Additionally, a high concentration of the porphyrin ring is itself toxic to many bacteria [3]. Heme can partition into the lipid bilayer and disrupt normal phospholipid bilayer function resulting in cell leakage and lysis [4], and can also catalyze the formation of highly reactive alkoxyl and peroxyl radicals that damage biomolecules including lipids, proteins, and DNA.

Male and female tsetse flies (Diptera: Glossinidae) both feed exclusively on vertebrate blood. Tsetse can house several microorganisms that may include pathogenic African trypanosomes (the etiological agents of human and animal African trypanosomiases), viruses, and symbiotic bacteria [5]. Heme liberated from digestion of the large blood meal (2–3 times the fly's body mass each feeding) presents a metabolic challenge for the fly and its microbial partners. One of tsetse's symbiotic bacteria, *Sodalis glossinidius*, resides intra- and extracellularly

within tsetse's midgut [6, 7]. Extracellular *S. glossinidius* are exposed to heme when its tsetse host consumes vertebrate blood. The bacterium contains a functional heme import system composed of the outer membrane protein HemR and a periplasmic/inner membrane ABC heme permease system (HemTUV) for use of the heme as an iron source [8]. However, nothing is known about how *S. glossinidius* tolerates and detoxifies heme, nor why the bacterium maintains genes necessary for heme biosynthesis [9]. As a first step in elucidating these mechanisms, we identify *Sodalis* genes that are differentially expressed in the presence of high heme, and then monitor *Sodalis'* ability to reside in tsetse's midgut when expression of a selection of these genes is experimentally eliminated by mutagenesis. Our results provide insight into how a symbiotic bacterium that resides in the gut of a hematophagous arthropod survives when exposed to high quantities of heme. This work may have epidemiological implications, as enteric symbiotic bacteria are well known modulators of pathogen infection establishment and transmission processes in several arthropod vector model systems.

## Methods

Insect maintenance. *Glossina morsitans morsitans* (*Gmm*) were maintained in the Yale School of Public Health insectary at 25°C with 60–70% relative humidity. All flies received defibrinated bovine blood (Hemostat Laboratories) every 48 hours through an artificial membrane feeding system [10]. All experimentally derived changes to this feeding scheme are detailed in their respective Materials and Methods sections below.

### Bacterial strains, plasmids, and growth conditions

Bacterial strains and plasmids used in this study are listed in Table 1. *E. coli* strains were grown in Luria-Bertani Broth (LB) with aeration or on Luria-Bertani Agar (L Agar) plates. All *E. coli* cultures were incubated at 37°C. *Sodalis glossinidius morsitans* (hereafter designated *Sgm*) were obtained by homogenizing a two week old *Gmm* pupae in 100 µl of Brain Heart Infusion (BHI) broth and then plating the homogenate on BHI agar plates supplemented with 10% defibrinated bovine blood (BHIB). A single colony was isolated and designated $Sgm^F$. Plates were incubated at 25°C and 10% $CO_2$. *Sgm* clones were harvested and grown in BHI at 25°C and 10% $CO_2$ in petri dishes without aeration.

Antibiotics were used for *E. coli* at the following concentrations: carbenicillin (carb) 125 µg/ml, ampicillin (amp) 50 µg/ml, chloramphenicol (cam) 30 µg/ml, and kanamycin (kan) 50 µg/ml. Antibiotics were used for *Sgm* at the following concentrations: carbenicillin (carb) 125 µg/ml, chloramphenicol (cam) 3 µg/ml, and kanamycin (kan) 25 µg/ml.

### Bacterial growth assays in heme

Clonal populations of *Sgm* were subcultured into BHI broth or BHI broth supplemented with hemin (0–150 µM, hereafter referred to as heme) and incubated at 25°C and 10% $CO_2$. Growth was measured by $OD_{600}$.

### RNA sequencing and data analysis

*Sgm* cultured under normal (BHI) and high heme conditions (BHI supplemented with 100 µM heme for 24 hours) were used as control and treatment samples, respectively, for RNA-seq analyses. Total RNA was extracted from three distinct clonal populations of treatment and control cells ($5 \times 10^8$ cells per replicate) using TRIzol reagent according to the manufacturer's (Invitrogen) protocol. Purified RNA was subjected to DNase treatment using the TURBO DNA-free kit (Ambion) and quality checked on an Agilent 2100 Bioanalyzer RNA Nano chip.

**Table 1. Bacterial strains and plasmids.**

| Strain or Plasmid | Characteristics | Reference |
|---|---|---|
| Bacterial Strains | | |
| *E. coli* Strains | | |
| DH5α | *endA1 hsdR17 supE44 thi-1 recA1 gyrA relA1* Δ(*lacZYA-argF*)*U169 deoR* [Φ80*dlac*Δ(*lacZ*)*M15*] | [11] |
| BL21(DE3) | F⁻*omp*T *hsd*S_B (r_B⁻, m_B⁻) *gal dcm* (DE3) | Novagen |
| *Sodalis* Strains | | |
| Sgm^F | *S. glossindius* from *Glossina moristans moristans* | S. Aksoy |
| *Sgm*^F-PAR | Parent strain for Sgm intron mutagenesis (Sgm^F/pAR1219) | This study |
| URSOD7 | Kan^R intron insertion at nucleotide 25 of SG1505 | This study |
| URSOD25 | Kan^R intron insertion at nucleotide 132 of SG0074 | This study |
| URSOD26 | Kan^R intron insertion at nucleotide 180 of SG2427 | This study |
| URSOD27 | Kan^R intron insertion at nucleotide 345 of SG2179 | This study |
| URSOD28 | Kan^R intron insertion at nucleotide 126 SG2061 | This study |
| URSOD29 | Kan^R intron insertion at nucleotide 390 of SGP2_0009 | This study |
| URSOD31 | Kan^R intron insertion at nucleotide 24 of SGP1_0002 | This study |
| URSOD32 | Kan^R intron insertion at nucleotide 114 of SG0437 | This study |
| URSOD33 | Kan^R intron insertion at nucleotide 33 of SG1100 | This study |
| URSOD35 | Kan^R intron insertion at nucleotide 348 of SG0182 | This study |
| URSOD40 | Kan^R intron insertion at nucleotide 102 of SG2280 | This study |
| URSOD41 | Kan^R intron insertion at nucleotide 99 of SG1275 | This study |
| Plasmids | | |
| pAR1219 | T7 polymerase under control of *lac* UV5 promoter for inducing intron mutagenesis | [12] |
| pET-22b | Expression vector; carbenicillin resistance | Novagen |

mRNA libraries were prepared using the Illumina Ribo-Zero rRNA Depletion Kit following the manufacturer's protocol. The six libraries were sequenced (paired-end) at the Yale Center for Genome Analysis using the Illumina HiSeq2500 system.

Using CLC Genomics Workbench version 8.5 (Qiagen), transcriptome reads were first trimmed and filtered to remove ambiguous nucleotides and low-quality sequences. The remaining reads were mapped to the annotated *Sgm* genome (bacteria.ensembl.org, genome ID GCA_000010085). Reads aligning uniquely to *Sgm* transcripts were used to calculate differential gene expression using CLC Genomics Workbench by employing a pairwise Baggeley's test, corrected with a False Discovery Rate (FDR) at $p<0.05$. For clusters of orthologous genes (COG) analysis, the COGs mapped to *Sgm* genes were downloaded from the Integrated Microbial Genomes and Microbiomes (IMG/M) site hosted by the Department of Energy's Joint Genome Institute (JGI) (img.jgi.doe.gov).

## Quantitation of *Sgm* gene expression *in vivo* in tsetse's midgut

Midguts dissected from female tsetse flies 24 hrs (considered recently replete) or 96 hrs (considered starved) after their last bloodmeal ($n$ = 6 biological replicates from each timepoint, three midguts per replicate) represented high heme and low heme environments, respectively. Total RNA was extracted using Trizol reagent (Invitrogen) and then treated with DNase (Turno DNA-free kit, Thermo Fisher) to remove contaminating DNA. cDNA was generated from 200 ng of total RNA purified from high and low heme midguts (using a Bio-Rad iScript cDNA synthesis kit) by priming the reaction with random hexamers to ensure the capture of *Sgm* derived transcripts. Primers used to amplify *Sgm* heme responsive cDNAs are listed in S1 Table. Constitutively expressed *Sgm rplB* was used to normalize transcript levels in each sample.

## Construction of *Sgm* mutants

Mutations in the heme-induced genes (HIGs) were constructed using the Targetron Intron Mutagenesis kit (Sigma-Aldridge, St. Louis, MO) as described previously [13]. Briefly, the group II intron on pACD4K-C or pACD4K-C-loxP was altered according to the manufacturer's instructions to contain a targeting site located within the coding sequence of the HIG. The altered intron plasmid containing the HIG targeting intron was electroporated into *Sgm*^F-PAR using the electroporation protocol describe by Dale et. al. [14]. pAR1219 carries the IPTG-inducible T7 RNA polymerase gene, which is required for transcription of the targeting intron in intron mutagenesis. Intron expression was induced with 500 μM isopropyl-β-D-thiogalacto-pyranoside (IPTG) for 1 hour and *Sgm* containing chromosomally inserted introns were selected for on BHIB agar containing kanamycin. Single colonies were restreaked after approximately 1 week on BHIB agar plates containing kanamycin. Insertion of the intron into the *Sgm* HIG, and elimination of the wildtype gene, was confirmed by PCR analysis using *Sgm* primer pairs that flank the HIG (S1 Table).

## Tsetse *per os* colonization assay

Either the *Sgm* parent strain *Sgm*^F-PAR (control) or one of the HIG mutant strains (treatment) was added to heat inactivated (HI; 56°C for 30 min) bovine blood [at a concentration of 500 colony-forming units (CFU)/ml] and provided to flies ($n$ = 25 per group) through an artificial membrane system [10]. Following *per os* inoculation with bacteria, flies were maintained on HI blood every 48 hr. Gut tissues were microscopically dissected at 1, 5 and 10 days post-inoculation with bacteria. Guts from each group ($n$ = 5) and time point were homogenized in 0.85% NaCl, serially diluted, and plated on BHIB supplemented with kanamycin (50 μg/ml). CFU per plate were manually counted to determine the number of treatment or control *Sgm* residing in each fly midgut. All *per os* colonization assays were performed in duplicate.

# Results

## *Sgm* grows in high heme conditions

Because *Sgm* resides within the midgut of hematophagous tsetse flies, we hypothesized that the bacteria would be able to grow in media containing high levels of heme. To test this hypothesis, we compared the minimal inhibitory concentration (MIC) of heme for *Sgm*^F-PAR and *E. coli* BL21(DE3)/pET22b grown *in vitro* in broth culture. *Sgm*^F-PAR was able to grow in a significantly higher concentration of heme as compared to *E. coli* (Fig 1). The MIC of heme for *Sgm* grown *in vitro* in BHI at 25°C is between 100 and 150 μM. This is higher than the MIC for many other microbes, where the observed inhibitory concentration for heme is in the low μM

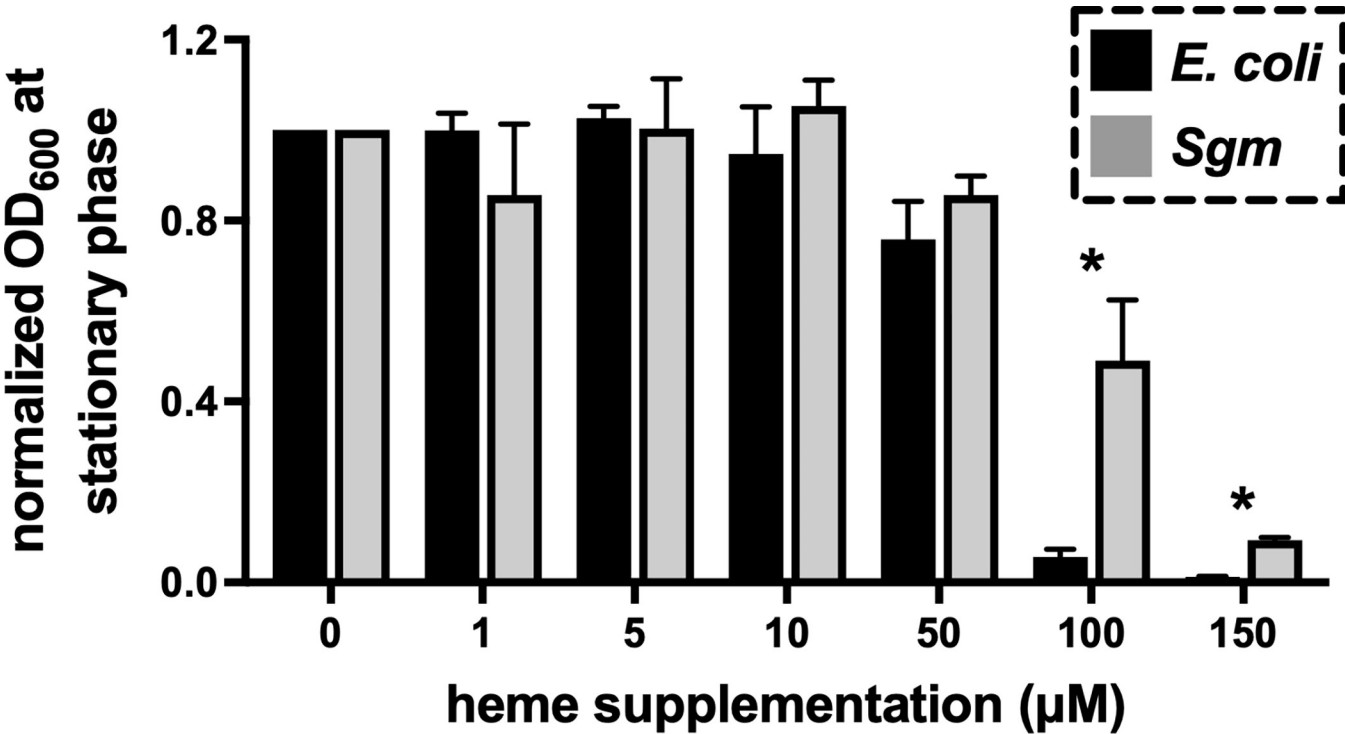

**Fig 1. Inhibition of bacterial growth by heme.** *E. coli* BL21(DE3)/pET22b and *Sgm*<sup>F-PAR</sup> cultures were inoculated at an optical density at 600 nm ($OD_{600}$) of 0.04–0.08 into their respective growth media (LB media for *E. coli* and BHI and for *Sgm*) containing increasing concentrations of heme. *E. coli* (black bars) was incubated at 37˚C and *Sgm* (grey bars) was incubated at 25˚C. The $OD_{600}$ of the cultures was measured at stationary phase (96 hours for *Sgm* and 24 hours for *E. coli*), and bacterial density at each heme dose was normalized to the density of cultures grown in the absence of added heme (0 μM). The averages of three trials are shown, and the standard deviation of the means are indicated by the error bars. Asterisks above bars indicate that *E. coli* and *Sgm* reached significantly different densities ($p<0.05$) at that heme concentration. Statistical significance was determined via multiple t-tests (GraphPad Prism v.9.4.1).

range (reviewed in [15]). *Sgm's* high MIC for heme likely reflects the bacterium's evolution in the blood rich niche of tsetse's gut.

## Identification of heme regulated genes in cultured *Sgm*

In the presence of high levels of heme, *Sgm* may alter gene expression to homeostatically mitigate heme toxicity. To identify *Sgm* heme regulated genes, we grew the bacteria in either normal BHI or BHI supplemented with 100 μM heme, which hereafter we refer to as 'high heme'. Log-phase cells from both groups ($n$ = 3 clonal populations per group) were harvested and subjected to RNA-seq analysis to identify genes that showed either increased (heme-induced genes, HIGs) or decreased (heme-repressed genes, HRGs) expression following exposure to high heme conditions. The total number of reads generated in each biological replicate, and the number of reads per biological replicate that mapped uniquely to the *Sgm* genome, are detailed in S2 Table (raw RNA-seq data is archived under NCBI BioProject ID PRJNA818891). Expression of 1495 genes was significantly differentially expressed in response to exposure to high heme (S1 Fig.). Of those, 219 genes were significantly induced at least 2-fold in the presence of high levels of heme, and a similar number of genes (217) were significantly repressed at least 2-fold (S2 Table). These 436 genes correspond to 18% of all genes in the *Sgm* genome. Many of the HIGs encode hypothetical proteins with unknown functions (50% of the heme-induced genes). Furthermore, a large percentage of HIGs encode genes involved in transcription (13%), replication/repair of DNA (12%), and carbohydrate transport

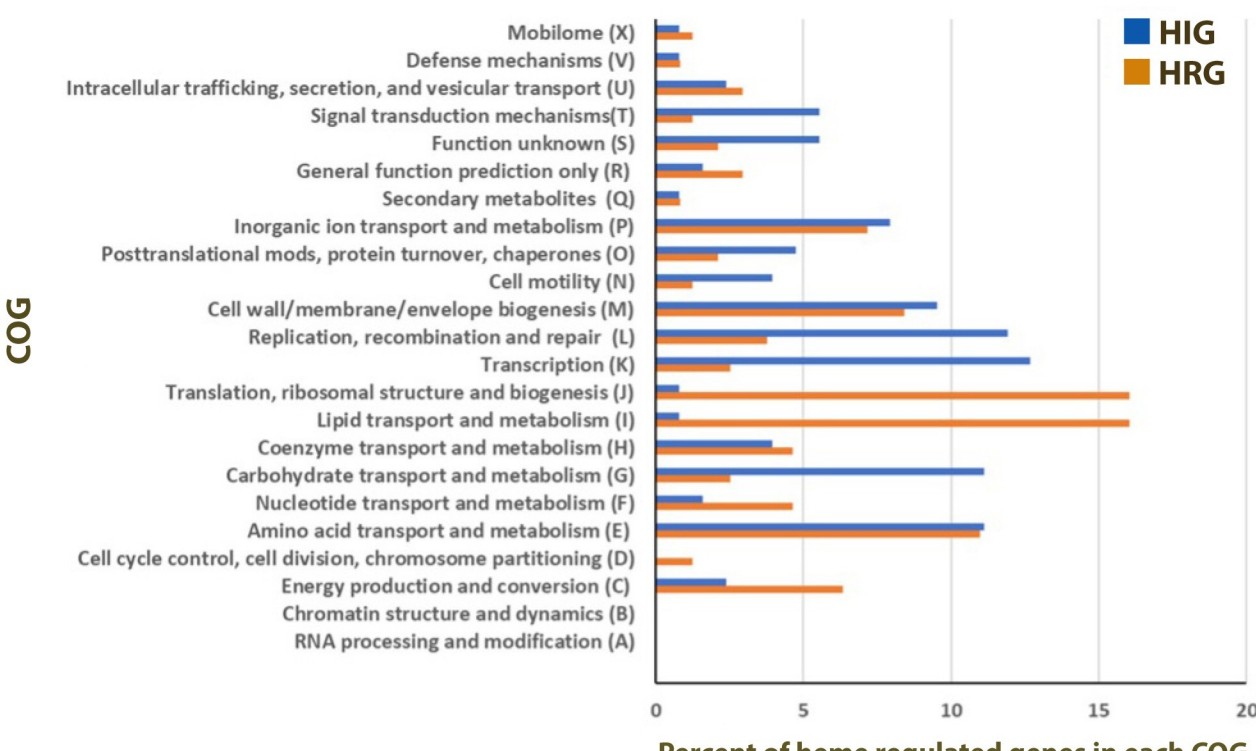

**Fig 2. Distribution of heme-regulated genes in COG categories.** *Sgm* genes that showed ≥ 2-fold change in expression when grown in the presence of high heme (100 μM) were grouped according to their clusters of orthologous group (COG) designations. HIG, heme-induced genes; HRG, heme-repressed genes.

and metabolism (11%) (Fig 2). The 20 genes that exhibited the largest changes in transcript abundance in response to high levels of heme are listed in Tables 2 and 3.

### High heme alters expression of *Sgm* heme and iron metabolism genes

*Sgm's* chromosome contains genes that encode two putative ferritins (SG2280 and SG1275) and a protein annotated as a bacterioferritin co-migratory protein (SG1728), all of which could store free iron that might be liberated from excess heme. All three genes exhibited modest (~1.8-fold) but statistically significant increases in expression in the high heme growth conditions. Furthermore, expression of gene SG1505, which encodes a putative heme binding protein that might sequester heme, was also heme induced (1.9-fold). In contrast, expression of *hemT*, *hemU*, and *hemV* genes, which encode a previously characterized inner membrane heme transport system in *Sgm*, was decreased 1.4-, 1.9-, and 1.8-fold, respectively, in high heme growth conditions [8]. Additionally, genes that encode proteins homologous to the Dpp dipeptide/heme ABC inner membrane transport system in *E. coli* (SG0061-SG0064) had decreased expression (1.5–2.7 fold) in high heme conditions.

### High heme alters expression of *Sgm* carbohydrate transport and metabolism genes

For the HIGs that had assigned COG designations, 11% of the heme-induced genes were associated with the carbohydrate transport and metabolism COG class G. These included genes annotated as part of putative L-fucose transport (SG0182-SG0183) and galactose ABC

**Table 2.** *Sgm* heme-induced genes (top 20).

| Gene | Putative Function | Induction level[b] (fold increase) |
|---|---|---|
| SG2427[a] | phosphotransfer system IIA component | 10.8 |
| SG2043 | hypothetical protein | 8.2 |
| SG2044 | hypothetical protein (possible transcriptional regulator) | 7.4 |
| SG1733 | hypothetical protein (possibly unique to *S. glossinidius*) | 7.3 |
| SG1832 | exisionase | 6.3 |
| SG2061[a] | anti-sigma28 factor FlgM (FlgM) | 6.1 |
| SG2113 | ornithine carbamoyltransferase chain I | 6.0 |
| SGP1_0054 | hypothetical protein | 5.7 |
| SG2396 | acetolactate synthase isozyme II small subunit | 5.4 |
| SG2179[a] | regulation of response to periplasmic stress (CpxP) homologue) | 5.1 |
| SG0437[a] | 2-isopropylmalate synthase (LeuA) | 5.1 |
| SGP1_0010 | hypothetical protein with peptidase domain | 4.8 |
| SG0944 | hypothetical protein | 4.7 |
| SG2049 | ipaD family Type 3 secretion system effector protein | 4.4 |
| SGP2_0009[a] | conserved hypothetical protein | 4.4 |
| SG1035 | phage integrase | 4.4 |
| SG0182[a] | fucose transport protein (FucP) | 4.4 |
| SGP1_0047 | transposase | 4.4 |
| SG2106 | transcriptional regulator | 4.2 |
| SG0148 | type 6 secretion system effector protein | 4.2 |
| SGP1_0002[a] | DNA-binding protein | 4.1 |

[a]HIGs selected for further analysis (*in vivo* expression and *in vivo* colonization phenotype of associated mutant) along with SG0074 (universal stress protein A, UspA), SG1100 (cold-shock DNA binding protein), SG1275 (ferritin, FtnA), SG2280 (bacterioferritin, Bfr), SG1505 (heme binding and/or degradation protein, HemS).

[b]Level of induction = fold-increase in gene expression in the presence of high versus low heme.

transport (SG0963-SG0965) systems. Furthermore, the HIGs also encode proteins that are part of three separate putative phosphotransferase systems (PTS) for sugar transport, including SG2041-SG2042 (PTS system component in the L-Ascorbate family), SG1327 (the first gene in an operon for a PTS system in the D-mannose family), and SG2427-SG2428 (PTS system, unassigned family). Finally, *SG1701*, which encodes a homologue of PtsH that is a common component in all PTS sugar transport systems, was induced 2-fold.

## Blood feeding alters the expression of *Sgm* heme regulated genes *in vivo* in tsetse's midgut

We next set out to determine whether *Sgm* genes that exhibited heme induced changes in expression *in vitro* similarly changed following exposure to high heme conditions *in vivo* in tsetse's midgut. We did so by quantifying the expression of a random sampling (across COG categories) of seven HIGs from Table 2 and one HRG from Table 3 in *Sgm* residing naturally in tsetse's midgut 24 hr (high heme environment) or 96 hr post-feeding (low heme environment) after flies received their last blood meal. We also quantified the expression of five additional HIGs (SG0074, SG1100, SG2280, SG1275, and SG1505) because they encoded potentially interesting proteins with respect to heme tolerance phenotypes (stress resistance and iron storage) (a summary of the *Sgm* genes we selected for in vivo expression analysis is shown in S3 Table). We observed that for each of the *Sgm* genes *SG0437*, *SG2427*, *SG1100*, *SPG1_0002*, *SG1275*, *SG0182*, *SG2179*, *SG2061*, *SG2280*, *SG1505*, and *SG1621*, the pattern of

**Table 3.** *Sodalis* repressed induced genes (top 20).

| Gene | Putative Function | Repression level[b] (fold decrease) |
|---|---|---|
| SG0680 | putative ammonium transport protein | 6.8 |
| SG0679 | nitrogen regulatory protein P-II | 6.8 |
| SG0253 | 30S ribosomal protein S21 | 6.5 |
| SG0620 | hypothetical protein | 5.6 |
| SG1976 | hypothetical protein | 5.0 |
| SG1621[a] | erythronate-4-phosphate dehydrogenase | 4.6 |
| SG1909 | thioredoxin 2 | 4.2 |
| SG1845 | conserved hypothetical protein | 4.0 |
| SG2279 | 30S ribosomal protein S10 | 4.0 |
| SG0645 | S-adenosylmethionine | 4.0 |
| SG0374 | protein-export protein SecG | 3.8 |
| SG0466 | pyruvate dehydrogenase complex repressor | 3.7 |
| SG1094 | arginine ABC transporter permease component | 3.5 |
| SG0468 | pyruvate dehydrogenase dihydrolipoyltransacetylase component | 3.5 |
| SG2277 | 50S ribosomal protein L4 | 3.5 |
| SG0256 | dihydroneopterin aldolase | 3.5 |
| SG2341 | magnesium transport protein | 3.4 |
| SG1202 | hypothetical phage protein | 3.4 |
| SG1748 | GMP synthase | 3.4 |
| SG2268 | 50S ribosomal protein L14 | 3.4 |

[a]HRG selected for *in vivo* expression analysis.

[b]Repression level = fold-reduction in gene expression in the presence of high versus low heme.

gene expression was similar both *in vitro* and *in vivo* in tsetse's midgut. Conversely, expression of *SGP2_0009* was undetectable in tsetse's midgut, and *SG0074* did not exhibit a significant change in expression in that environment (Fig 3). These results suggest that *Sgm* heme-responsive gene expression exhibits similar patterns in high heme conditions both *in vitro* and *in vivo* in tsetse's midgut. Additionally, the heme induced genes may play a role in *Sgm's* ability to tolerate and successfully reside within the high heme environment present in tsetse's midgut following consumption of a blood meal.

## Mutations in specific *Sgm* HIGs impact the bacterium's ability to grow and survive within tsetse's midgut

We determined that high heme conditions influence *Sgm* gene expression when the bacterium resides in tsetse's midgut. We thus tested whether any of the HIGs encode proteins that contribute to *Sgm's* ability to successfully colonize this niche. To do so, we used intron mutagenesis to generate 12 *Sgm* mutant strains, each of which fails to express one HIG, and then inoculated distinct groups of tsetse flies *per os* with either one *Sgm* HIG mutant strain or parent strain *Sgm*[F-PAR] (wildtype for each gene). We then monitored the density of HIG mutant strains (normalized to parent strain *Sgm*[F-PAR]) in tsetse guts at 1, 5, and 10 day time points post-inoculation. We observed that *Sgm* strains unable to express two putative sugar transporters [the PTS transporter component (SG2427) or the fructose transporter (SG0182)], bacterioferritin (SG2280), and a putative DNA binding protein (SGP1-0002), exhibited significant defects in survival and/or growth within tsetse's midgut (Fig 4). As such, these HIGs encode proteins that mediate *Sgm's* ability to reside within tsetse's gut where the environment contains abundant blood meal-derived heme.

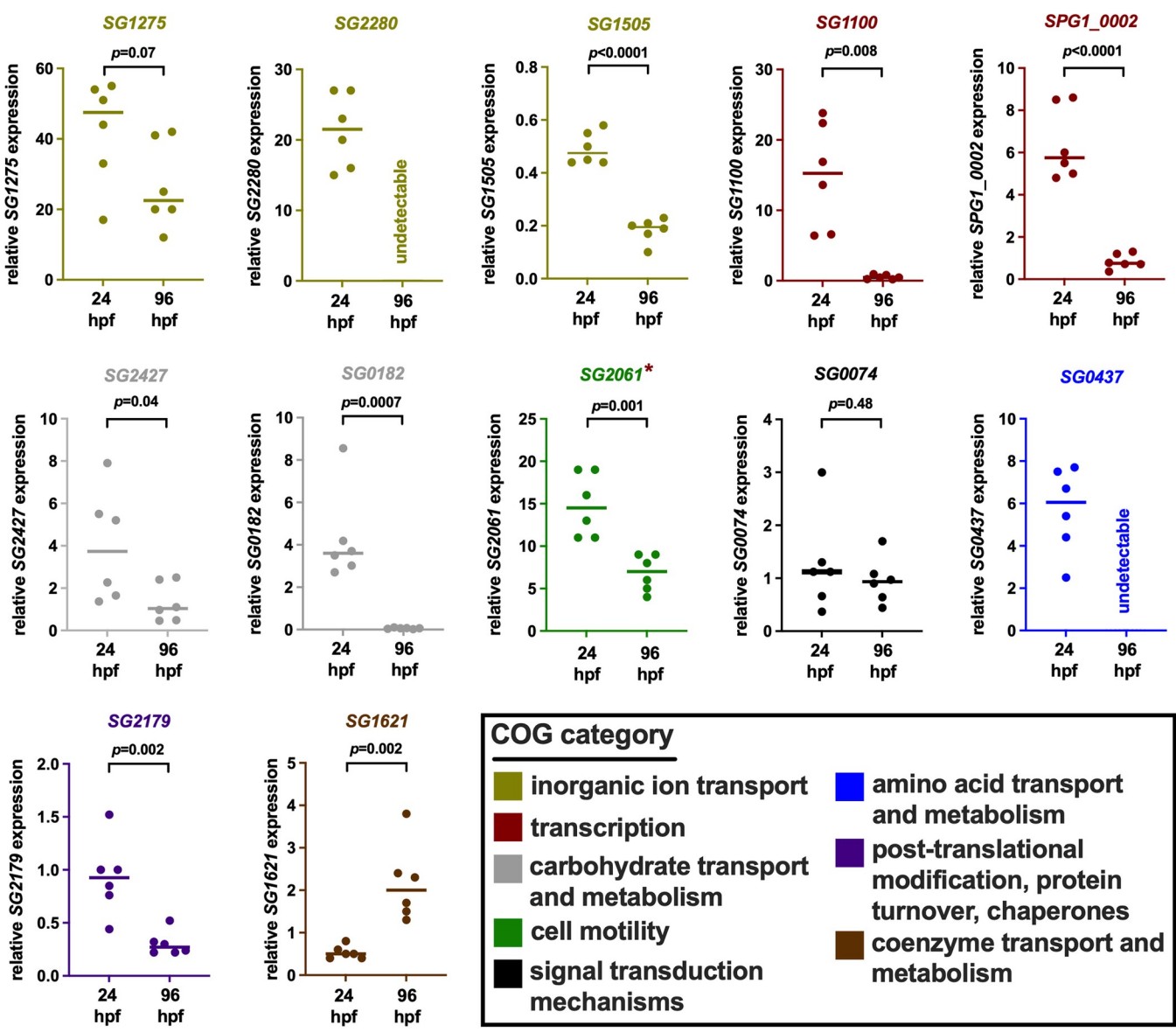

**Fig 3. Relative expression of *Sgm* heme-regulated genes *in vivo* in tsetse's midgut either 24 hr or 96 hr (high and low heme environments, respectively) after flies had received their last blood meal.** Expression levels of each gene were quantified by RT-qPCR and normalized to *Sgm's* constitutively expressed *rplB* gene. Individual dots represent one biological replicate, each containing three midguts. *Sgm* gene IDs and graph colors correspond to the COG category (shown in the box at bottom right) in which they each group. *Sgm* gene *SG2061* is indicated by a cayenne asterisks because it also groups within the 'transcription' COG. Expression of *SGP2_0009* was undetectable in *Sgm* that reside in tsetse's midgut and thus left out of this figure. hpf, hours post-feeding. Bars represent median values, and statistical significance was determined via Student's t-test (GraphPad Prism v.9.4.1).

## The predicted protein products of two genes (*SG0182* and *SG2427*) that enhance *Sgm* colonization of tsetse are homologous to sugar transporters

Because *SG0182* and *SG2427* mutants presented defects in tsetse fly colonization, we did additional bioinformatic analysis of the proteins encoded by these genes. We first compared the putative amino acid sequence of SG0182 with the nonredundant protein database using BLASTP. The putative protein is homologous to L-fucose:H+symport permeases in the Major Facilitator Superfamily in a wide variety of other bacterial species, and is 75% identical and 84% similar to the protein FucP, which transports fucose into *E. coli*. SG2427 is homologous to

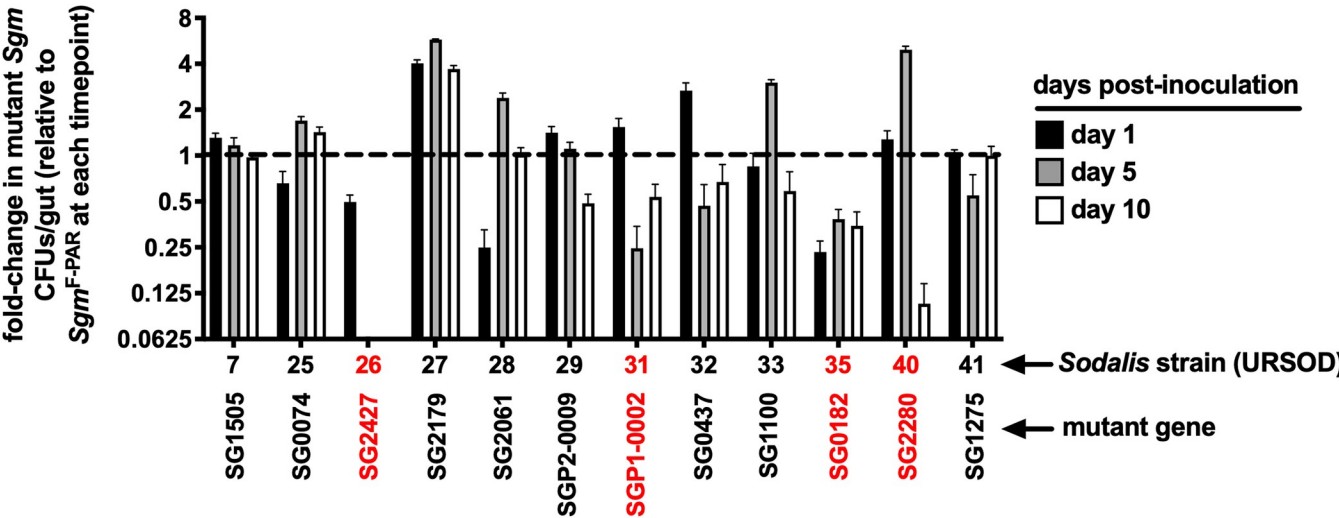

**Fig 4. Colonization of tsetse with *Sgm* HIG mutants.** Distinct groups of tsetse flies (*n* = 25 per group) were inoculated *per os* (500 CFU/ml of blood) with one *Sgm* mutant strain or the parent strain (*Sgm*^F-PAR^). Midguts (*n* = 5 per *Sgm* mutant and parent strain) were harvested at 1, 5, and 10 days post-inoculation and plated on selective media (kanamycin, 50 μg/ml) to determine bacterial density (CFUs/gut). Bars represent the fold-change in the number of CFUs/gut of each *Sgm* mutant strain relative to the number of CFUs/gut of the *Sgm* parent strain *Sgm*^F-PAR^ (mutant strain CFUs/parent strain CFUs) at each timepoint. The dashed line represents the value at which CFU/gut of parent strain *Sgm*^F-PAR^ and each mutant strain are the same. Strains indicated in red represent those that presented colonization defects in tsetse's gut. Mutant strains 26, 31, 35, and 40 grew to a significantly lower density, in tsetse's midgut compared to the parent strain by day 10 (*p* values for strains 26, 31, 35, and 40 = <0.0001, 0.003, 0.001, and <0.001, respectively). Statistical significance was determined via 2-way ANOVA with Tukey's multiple comparisons (GraphPad Prism v.9.4.1).

IIA subunits of the PTS sugar system in a wide variety of other bacterial species. In PTS systems, the phosphoryl group from phosphoenolpyruvate (PEP) is transferred to a relay of several proteins or protein domains (including PTS enzymes I, HPr, IIA, and IIB), which ultimately results in the phosphorylating of sugars during transmembrane enzyme IIC-mediated transport into the cell. Except for Enzyme I and HPr, all of the enzymes are specific for a particular class of sugars. The top BLASTP matches by E-value for the *SG2427* encoded protein were to putative PTS fructose transporter IIA subunits in *Sodalis praecaptivus*, *Raoultella terrigena*, *Enterobacter sp. 10–1*, *Superficieibacter electus*, and *Klebsiella pneumoniae*. SG2427 contains the Pfam domain (EIIA-man/PF03610) corresponding to the PTS system fructose IIA component.

Because PTS systems have been studied extensively in *E. coli*, and because *Sgm* and *E. coli* are in the same phylogenetic order (*Enterobacteriales*), we compared the putative SG2427 amino acid sequence to the proteins encoded by the *E. coli* K-12 substrain MG1655. SG2427 was 32% identical and 59% similar to the amino residues 14–128 of the 323 amino acid protein ManX, which is a PTS enzyme IIAB fusion protein that is part of the fairly promiscuous PTS system for transport of glucose, glucosamine, fructose, mannose, N-acetyl-glucosamine (Fig 5). Furthermore, SG2427 was 22% identical and 41% similar to putative PTS enzyme IIA component YadI in *E.coli*.

Like other PTS subunits, *SG2427* is the first gene in an operon with genes that encode other subunits of the PTS transport system. The putative proteins encoded by those two genes (*SG2426* and *SG2425*) were compared with the database of *E. coli* K12 proteins (Fig 5). SG2426 (putative PTS enzyme IIB component) was 25% identical and 52% similar to the amino residues 164–298 of the 323 amino acid fusion protein ManX (for transport of glucose, glucosamine, fructose, mannose, N-acetyl-glucosamine), 21% identical and 44% similar to PTS enzyme IIB component AgaV (for transport of N-acetyl-galactosamine), and 27% identical and 48% similar to PTS enzyme IIB component AgaB (transport of galactosamine). SG2425

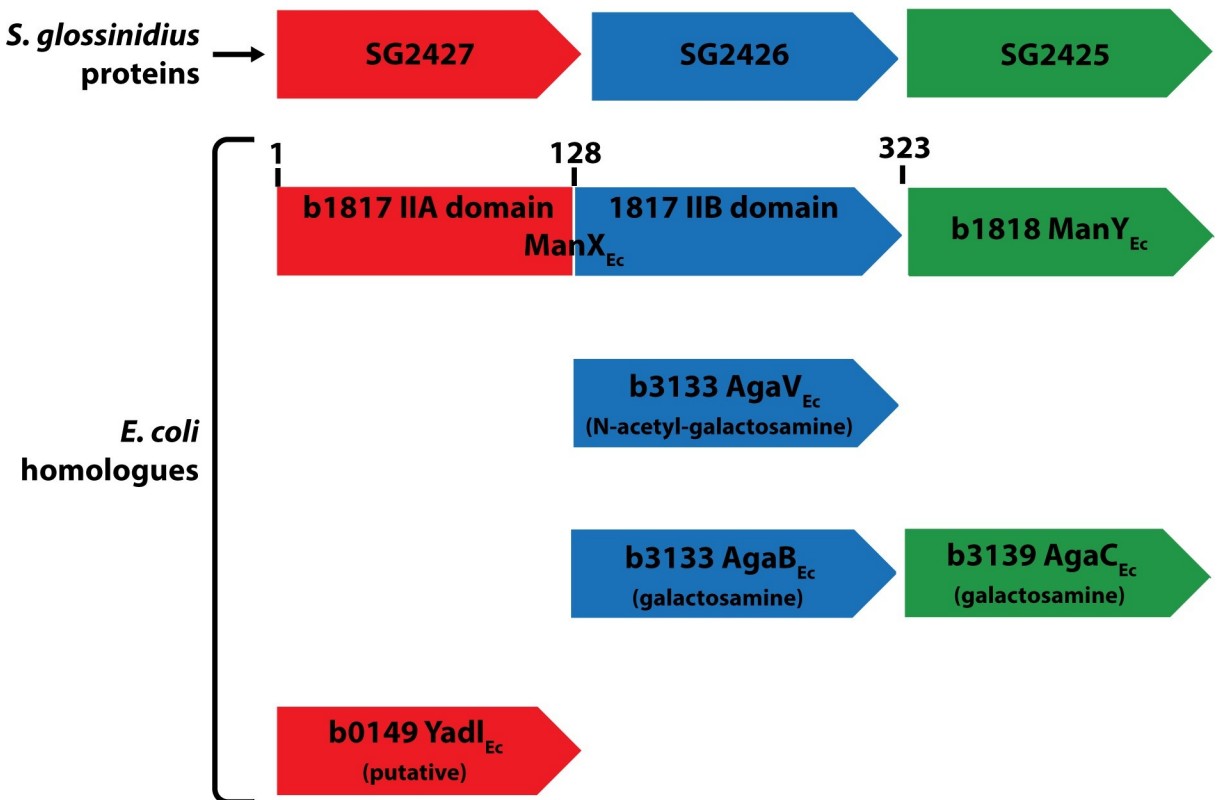

**Fig 5. Comparison of *Sgm* putative PTS proteins encoded by SG2425-2427 with *E. coli* K12 proteins.** The three putative *S. glossinidius* proteins are shown at the top in red, blue, and green, with their *E. coli* K12 homologues shown below in matched color coding.

(putative PTS enzyme IIC component) was 28% identical and 51% similar to PTS enzyme IIC component AgaC (for transport of galactosamine), and 26% identical and 51% similar to PTS enzyme IIC ManY (for transport of glucose, glucosamine, fructose, mannose, N-acetyl-glucosamine). The conservation of putative sugar transporter proteins in the *Sgm* genome, coupled with the defects in tsetse fly colonization of *Sgm* with mutations in the genes encoding these proteins, suggest sugar transporters play a key role in the ability of *Sgm* to mitigate heme toxicity.

## Discussion

Selective pressures associated with surviving blood meal processing have likely resulted in a suite of heme tolerance mechanisms in bacterial symbionts that reside in the hostile gut of hematophagous insects. In this context, the goal of this study was to acquire insight into how *Sgm*, an endosymbiont of the obligately hematophagous tsetse fly, survives in the high heme environment of its insect host's gut. We did so by culturing *Sgm* under high (100 μM of exogenous heme) or low (no exogenous heme added) heme conditions and then comparing the global transcriptional response between the two groups. We observed that approximately 200 *Sgm* genes were significantly induced at least 2-fold in response to 100 μM heme. A large percentage of these heme-induced genes mapped to orthologues in the 'transcription' COG category, suggesting complex transcriptional networks aid in adaptation to this environment by fine tuning gene expression including significant alteration of expression of transcriptional regulators in *Sgm*. Other bacterial transcriptomes show similar patterns of inducing

transcription-related functions in response to blood or heme [16–18]. Importantly, we also observed that a selection of *Sgm* genes induced in response to high heme culture conditions were similarly induced in *Sgm* that resided within the gut of replete as opposed to starved tsetse flies. Finally, *Sgm* strains with mutations in four HIGs (SG2427, SG0182, SG2280, and SGP1-0002) presented colonization defects when inoculated *per os* into tsetse flies. Our results taken together indicate that *Sgm* presents a robust transcriptional response when grown in high heme conditions, and that this genetic response is important for the bacterium to survive in the gut its tsetse host.

Processes that mitigate heme toxicity are predicted to be physiologically critical for *Sgm* that reside within the midgut lumen of hematophagous tsetse flies. As such, multiple tolerance mechanisms have likely evolved in the bacterium, including heme-responsive changes in gene expression that result in altered transport of heme across the bacterial cell membrane and/or intracellular sequestration of excess heme and iron. Our data suggest that *Sgm* reduces heme transport into the cell by decreasing expression of the genes *SG1538-1540* and *SG0061-SG0064*, encoding the HemTUV and Dpp dipeptide/heme ABC inner membrane transport systems. We also found that transcription of *Sgm SG1505* increases in response to high heme levels *in vitro* and is also induced in enteric *Sgm* after the fly host consumes a blood meal. This gene encodes HemS, which shares significant homology with proteins predicted to bind heme, thus reducing its toxicity, and transfer heme to other heme-degrading proteins [19, 20]. Furthermore, the HemS homologue in *Yersinia pseudotuberculosis* (HmuS) has heme oxygenase activity that cleaves the protoporphyrin ring to release iron from heme [21]. While HemS may contribute to protecting *Sgm* from excess heme-associated toxicity, the protein is not essential as SG1505 *Sgm* mutants grow in high heme media and colonize tsetse at levels comparable to the parent strain. This could also reflect the fact that tsetse's genome encodes a heme oxygenase [22, 23]. We also found that the expression of genes predicted to encode ferritin or ferritin-related proteins, which exhibit iron-storage functions in other bacteria, was induced by heme and induced in the fly after a blood meal. Mutagenesis of *Sgm* SG2280, which encodes a putative bacterioferritin, reduced the colonization of flies, suggesting that modulating iron levels is important *in vivo*.

Our data indicates that *Sgm* presents a general stress response when exposed to high heme. Seventy-eight percent of the genes encoding ribosomal proteins exhibit a modest (average of 2-fold) decrease in expression. *SG1485* and *SG0074*, which encode homologues of the bacterial universal stress proteins UspE and UspA, respectively, present 2- and 3-fold changes in expression [24, 25]. These proteins mediate responses for several different stressors, including oxidative stress, acid stress, and growth arrest, and have been implicated in facilitating colonization of bacterial pathogens (reviewed in [26, 27]). *Sgm SGP1-0002* presented a 4-fold increase in transcript abundance under high heme conditions, and the *SGP1-0002* HIG mutant exhibited a colonization defect in tsetse's gut. *SGP1-0002* encodes a putative DNA binding protein that is 41% and 48% identical to paralogous H-NS and StpA global transcriptional regulators in *E. coli*. H-NS functions as both a scaffold protein that contributes to the nucleoid structure and as a global regulator of gene expression [28]. Additionally, stress resistance proteins are regulated by homologues of H-NS in several bacterial species [29, 30]. In one case, the *Porphyromonas gingivalis* DNA-binding protein PgDps is required for growth of the bacterium in high heme conditions and protects the bacterium from oxidative stress [31]. Our results suggest that maintenance of *Sgm* homeostasis during heme stress is mediated by a significant change in the expression of transcription and translation regulators. More work is required to determine if *Sgm* that reside within tsetse's midgut respond similarly to high heme conditions.

Free iron released from heme can catalyze the formation of damaging hydroxyl radicals via Fenton chemistry or lipid peroxidation [32]. This process may result in the upregulation of

bacterial oxidative stress detoxification genes in high heme environments. In *Sgm*, three genes known to counteract ROS were heme induced: *SG0017* (encodes a putative manganese superoxide dismutase, which is already highly expressed by *Sgm* grown in BHI and induced 1.5-fold with the addition of heme), *SG2325* (Fe/S biogenesis), and *SG2047* (putative organic hydroperoxide resistance regulator). Other *Sgm* genes identified by Pontes et. al. [33] as potentially mitigating oxidative stress survival were not heme-induced in this study. This pattern of gene expression may be interpreted in different ways. In one scenario, the oxidative stress burden on *Sgm* cultured in the presence of 100 μM heme may actually be lower than predicted because the bacterium could employ other mitigation strategies that quickly lower ROS. Alternatively, *Sgm's* response to high heme *in vitro* may reflect how the bacterium copes with this challenge when it resides naturally in tsetse's midgut. In this scenario, *Sgm* may utilize its own oxidative stress-reducing mechanisms as well as benefit from those that its tsetse fly host employs to overcome excessive heme-induced ROS. In fact, a precedent for this theory exists in the mosquito *Aedes aegypti*, where immediately following a blood meal, heme mediated activation of protein kinase C causes decreases in ROS in the gut [34]. In a third scenario, some of *Sgm's* oxidative stress survival genes might be constitutively expressed. In this case, the transcriptional response that coordinates changing the expression of a large number of oxidative stress survival genes in response to environmental signals may have been lost over the course of *Sgm's* evolution in tsetse. Consistent with this idea, of the two major ROS sensing transcriptional regulator systems found in gamma-proteobacteria (OxyR and SoxR/S), *Sgm* has retained only OxyR during reductive evolution. If ROS levels are consistently high in tsetse's gut, this may have selected for constitutive high-level expression of *Sgm* detoxification genes (i.e repressor proteins are not required to keep expression off at certain times). In fact, *Sgm* superoxide dismutase (SG0017) and a putative peroxidase (SG0642) are highly expressed in BHI, regardless of heme levels.

Eleven percent of the HIGs were in the carbohydrate transport and metabolism COG class. Regulation of carbohydrate transport and metabolism gene expression by exposure to heme, hemoglobin, or blood results in differential expression of genes associated with carbohydrate transport and metabolism in several bacterial pathogens [16,17,18,35,36]. In *Sgm*, the heme signal could correlate with the presence of erythrocytes whose membranes contain glycoproteins and serum sugars that could be used by the bacterium for growth. For example, fucose is found on the surface of erythrocytes [35], and *Sgm* SG0182, which encodes a putative fucose transporter FucP, was induced 4.4-fold in the presence of high heme. Deletion of this gene resulted in a slight decrease in *Sgm* colonization of tsetse, suggesting that the ability to import or bind to fucose might help *Sgm* colonize tsetse's gut. Additionally, *SG0963-SG0965*, which constitute an operon that encodes a putative galactose ABC transporter, were induced 2 to 3.7-fold by heme. Galactose present in the blood as a result of human digestion and absorption, and galactosylceramides have been detected on the membranes of both erythrocytes and leukocytes (reviewed [37]). Furthermore, the first gene in an operon that encodes the promiscuous PTS sugar transporter (*SG1327*, ManX), and *SG1701*, which encodes a putative homologue of a common component in all PTS sugar transport systems (PtsH), were induced in *Sgm* under high heme conditions [9, 38]. Finally, mutation of SG2427 (encodes a putative PTS sugar transporter) inhibited the ability of *Sgm* to colonize tsetse's midgut. However, the identity of the transported molecule is unknown. Collectively, these data suggest the presence of heme may signal the availability of blood sugars that may be important for *Sgm* growth in tsetse.

*Sgm* infection prevalence, and density of infection, can vary significantly within and between populations of wild tsetse flies [39–42]. Although the physiological mechanisms that underlie these differences in *Sgm* infection dynamics have never been experimentally

addressed, we speculate that heme tolerance could be one contributing factor. More specifically, different tsetse populations feed on distinct vertebrate hosts, which may present different concentrations of heme in their blood [likely based on their thermogenic properties [43] and/ or diet [44]]. Different tsetse populations could house *Sgm* that exhibit genetic variation within genes that mediate the bacterium's response to heme toxicity. These variable genotypes could account for population-dependent discrepancies in *Sgm* infection prevalence and infection density. Similarly, tsetse flies within different populations may present variability in genes responsible for mediating blood meal processing mechanisms [45]. If this were the case then the amount of gut-associated free heme could be tsetse population dependent, and thus in some cases not enough to be toxic to *Sodalis*. Importantly, the dynamics of *Sodalis* infection in tsetse is of epidemiological consequence, as flies that harbor the bacterium at high density in their midgut are more susceptible to infection with pathogenic African trypanosomes than are their counterparts that lack the bacterium or that harbor the bacterium at relatively low densities [40, 41, 46–48]. Thus, *Sgm* plays a prominent role in the parasite's ability to complete its tsetse-specific developmental program and be transmitted to a new vertebrate host. More broadly speaking, in addition to tsetse flies, mosquitoes [49, 50], ticks [51], and sandflies [52] also house vector competence-mediating symbionts that must be able to survive within the high heme environment of their host's gut. Our results, which provide insight into the physiological mechanisms that facilitate the retention of bacterial symbionts in the hostile gut environment of hematophagous arthropods, may contribute to the development of novel vector-borne disease control strategies.

## Supporting information

**S1 Fig. Volcano plot depicting global transcriptomic changes in *Sgm* cultured in BHI broth supplemented with 100 µM heme vs. normal BHI broth.** All *Sgm* genes detected by RNA-seq are plotted on the graph, and each dot represents one gene. Genes represented in red are significantly differentially expressed (*p*-value of $\leq 0.05$ and a fold-change $\geq 2$, indicated by dashed, grey lines) in treatment vs. control cells. Green and blue genes are listed in Table 2, and green genes represent those experimentally mutated in *Sgm* and assayed for their colonization phenotype in tsetse's gut (results shown in Fig 4).
(TIF)

**S1 Table. PCR primers used in this study.**
(DOCX)

**S2 Table. Raw transcriptome data from *Sodalis glossinidius* cultured in BHI containing 100 µM hemin for 24 hours compared to cells cultured in non-supplemented BHI.**
(XLSX)

**S3 Table. Phenotypes associated with selected Sgm heme-induced genes.**
(DOCX)

## Acknowledgments

We sincerely thank Dr. Serap Aksoy (Yale School of Public Health) for use of equipment in her laboratory.

## Author Contributions

**Conceptualization:** Laura J. Runyen-Janecky, Brian L. Weiss.

**Data curation:** Laura J. Runyen-Janecky, Aurelien Vigneron, Brian L. Weiss.

**Formal analysis:** Laura J. Runyen-Janecky, Aurelien Vigneron, Brian L. Weiss.

**Funding acquisition:** Laura J. Runyen-Janecky, Brian L. Weiss.

**Investigation:** Laura J. Runyen-Janecky, Jack D. Scheutzow, Ruhan Farsin, Leah F. Cabo, Katie E. Wall, Katrina M. Kuhn, Rashel Amador, Shaina J. D'Souza, Aurelien Vigneron, Brian L. Weiss.

**Methodology:** Laura J. Runyen-Janecky, Aurelien Vigneron, Brian L. Weiss.

**Project administration:** Laura J. Runyen-Janecky, Brian L. Weiss.

**Resources:** Laura J. Runyen-Janecky, Brian L. Weiss.

**Software:** Laura J. Runyen-Janecky, Aurelien Vigneron, Brian L. Weiss.

**Supervision:** Laura J. Runyen-Janecky, Brian L. Weiss.

**Validation:** Laura J. Runyen-Janecky, Aurelien Vigneron, Brian L. Weiss.

**Visualization:** Laura J. Runyen-Janecky, Aurelien Vigneron, Brian L. Weiss.

**Writing – original draft:** Laura J. Runyen-Janecky, Brian L. Weiss.

**Writing – review & editing:** Laura J. Runyen-Janecky, Katie E. Wall, Katrina M. Kuhn, Aurelien Vigneron, Brian L. Weiss.

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
