## [Decision Letter · Decision Letter 0]

24 Oct 2022

Dear Dr. Weiss,

Thank you very much for submitting your manuscript "Heme-induced genes facilitate endosymbiont (Sodalis glossinidius) colonization of the tsetse fly (Glossina morsitans) midgut" for consideration at PLOS Neglected Tropical Diseases. As with all papers reviewed by the journal, your manuscript was reviewed by members of the editorial board and by several independent reviewers. The reviewers appreciated the attention to an important topic. Based on the reviews, we are likely to accept this manuscript for publication, providing that you modify the manuscript according to the review recommendations. 

Sincerely,

José M. C. Ribeiro

Academic Editor

Mathieu Picardeau

Section Editor

Reviewer's Responses to Questions

**Key Review Criteria Required for Acceptance?**

**Methods**

-Are the objectives of the study clearly articulated with a clear testable hypothesis stated?

-Is the study design appropriate to address the stated objectives?

-Is the population clearly described and appropriate for the hypothesis being tested?

-Is the sample size sufficient to ensure adequate power to address the hypothesis being tested?

-Were correct statistical analysis used to support conclusions?

-Are there concerns about ethical or regulatory requirements being met?

Reviewer #1: Methods are well described and appropriate. Error bars are missing from one figure.

Reviewer #2: Yes for all. 

Essentially, I would like to see a comparison between the heme concentration chosen with another one below the toxicity threshold (which my “smart guess” says it is what I expect to find in the in vivo situation), to exclude the stress response component. However, this would be out of the scope of this manuscript. Please, take this as a suggestion for further research.

**Results**

-Does the analysis presented match the analysis plan?

-Are the results clearly and completely presented?

-Are the figures (Tables, Images) of sufficient quality for clarity?

Reviewer #1: Figure 2 is not legible.

Reviewer #2: About the results I have a couple of specific questions:

Line 279 -figure 3 – as the labile heme pool in the gut of glossina is not known (see comments on discussion), the term high and low heme is instead a hypothesis than a fact. So a change for something more attached to the experimental condition (just time after a blood meal) would be better.

Line 396 – if iron is important, then heme degradation should be modulated as well. Are there symbiont heme oxygenase homologs? Or is this ferritin important for handling non-heme iron? Or iron produced by a heme-degrading insect HO?

Is there a functional heme biosynthesis pathway in this bacteria? In the table, there are several genes related to achromobactin biosynthesis (however, I do not know if the complete route is found in Sgm genome), which is a siderophore, possibly related to iron sequestration, together with the role of the ferritin. This could mean that, in the presence of heme-derived stress, heme uptake is inhibited and iron sequestration is activated to provide a source of iron, or alternatively to simply work as an iron chelator. 

Why are there so many type III secretion apparatus proteins upregulated by heme? 

This symbiont works as a biotin source for the insect? Heme inducing activation of biotin biosynthesis would be an interesting aspect of the mutualistic interaction between host and bacteria.

**Conclusions**

-Are the conclusions supported by the data presented?

-Are the limitations of analysis clearly described?

-Do the authors discuss how these data can be helpful to advance our understanding of the topic under study?

-Is public health relevance addressed?

Reviewer #1: Yes, the main conclusions are all well supported.

Reviewer #2: Although several genes are clearly related to a heme response, and the authors provide evidence that, at least for some selected transcripts, a similar expression profile is also found in the insect midgut, my feeling is that they are dealing with a profile that has a component of a general stress response, as they see a reduction of major ribosomal components (they are abundant among the heme-repressed genes). Does this downregulation of protein synthesis also occur in vivo? Also, several stress response proteins are upregulated. 

The experimental design chosen was to use a heme concentration that is deleterious but not lethal. I can understand the rationale for that strategy. However: what are the “free” heme concentrations that are attained in the midgut in vivo? Is there an experimental evaluation of labile heme (an expression to describe the heme pool that is available to change from one binding partner to another)? As these data are probably lacking, at least, can the authors provide a description of the rate of blood protein degradation in Glossina? This might work as an indirect (but acceptable) evidence that labile heme is being generated. 

The common pattern from published research is that heme released from hemoglobin is quickly converted into aggregates in the gut of mosquitoes, kissing bugs, and ticks, but, as long as I know, there are not such studies available for Glossina. So, I wonder if heme (labile) concentrations are in fact escalating to 100micromolar. 

Also, Sodalis is not the unique bacteria found in the Glossina midgut. The use of the antibiotic selects the mutant strain and allows comparison in a situation where other species are not present. As a simplifying experimental approach, this is ok but is a fact that needs to be recognized in the discussion.

**Editorial and Data Presentation Modifications?**

Reviewer #1: (No Response)

Reviewer #2: (No Response)

**Summary and General Comments**

Reviewer #1: Runyen-Janecky et al report on an RNAseq analysis of genes regulated by heme exposure in the gut endosymbiont of tsetse. They hypothesized that as part of the process of adaptation to become a commensal in the gut of an obligatory bloodfeeder, the Sodalis bacteria should be able to tolerate extreme heme concentrations. Indeed, their data show this is the case and that many genes are differentially regulated in the bacteria in response to heme. Of these, a subset were mutagenized and shown to reduce colonization of the gut, demonstrating their adaptive nature. Overall, this is a well put together, original study that makes an important contribution to the field of vector biology. I will add that the manuscript is also extremely well written, and should be accessible to the broad audience at PLoS NTD. I only have a few minor concerns and some suggestions for data presentation that I hope the authors will consider.

In Figure 4, the authors show the results of colonization experiments with Sodalis strains deficient in various genes that had responded to heme. There are no error bars on the graph, even though the methods and legend indicate there were replicates. From the legend, N=25 per group, with n=5 for each data point. So should be five data points per time period. The graph itself is also odd. The Y-axis is labelled "log2 CFU/gut (Sgm-mutant/Sgm-WT)”, and this is fine. However, the numerical labels are not log2 values, they are linear values but on a log2 scaled axis. This is a bit confusing, as it would be simpler to just number from 1-4 (and -1 to-4), since those are the log2 values. 

While the authors include all of the expression data values in the supplement, there is no corresponding figure in the manuscript that summarizes these data. There are many acceptable ways of showing the data: 1) linear plot of expression heme vs no heme for all genes; 2) volcano plot showing log2 fold change vs corrected p-value; 3) log10 (heme x non-heme expression) vs log2 fold change. Any of these would help to demonstrate the strong assymetric shift in expression that occurs upon treatment in a way that is essentially overlooked without such a summary.

The summary in Fig 2 is not very effect in communicating categories. There are too many similar colors, and the fonts on the figure are too small to read. Recommend changing to a different format.

Reviewer #2: Overall, a very interesting paper, the first to study how a symbiont from the intestinal microbiota of a blood-sucking insect deal with heme. It creates a straightforward experimental approach using an obligate symbiont and combines a transcriptome and mutational strategy.

PLOS authors have the option to publish the peer review history of their article (what does this mean?). If published, this will include your full peer review and any attached files.

Reviewer #1: No

Reviewer #2: Yes: Pedro L Oliveira

Figure Files:

Data Requirements:

Reproducibility:

References

---

## [Editor Report · Decision Letter 1]

18 Nov 2022

Dear Dr. Weiss,

We are pleased to inform you that your manuscript 'Heme-induced genes facilitate endosymbiont (Sodalis glossinidius) colonization of the tsetse fly (Glossina morsitans) midgut' has been provisionally accepted for publication in PLOS Neglected Tropical Diseases.

Best regards,

José M. C. Ribeiro

Academic Editor

Mathieu Picardeau

Section Editor

---

## [Editor Report · Acceptance letter]

23 Nov 2022

Dear Dr. Weiss,

We are delighted to inform you that your manuscript, "Heme-induced genes facilitate endosymbiont (Sodalis glossinidius) colonization of the tsetse fly (Glossina morsitans) midgut," has been formally accepted for publication in PLOS Neglected Tropical Diseases.

Best regards,

Shaden Kamhawi

co-Editor-in-Chief

Paul Brindley

co-Editor-in-Chief
